# VSTM1-v2 does not drive human Th17 cell differentiation: A replication study

**Helen J. von Richthofen**[1,2☉], **Florianne M. J. Hafkamp**[3☉], **Anouk van Haperen**[1,2], **Esther C. de Jong**[3‡], **Linde Meyaard**[1,2‡]*

1 Center for Translational Immunology, University Medical Center Utrecht, Utrecht University, Utrecht, The Netherlands, 2 Oncode Institute, Utrecht, The Netherlands, 3 Department of Experimental Immunology, Amsterdam Institute for Infection & Immunity, Amsterdam University Medical Center, University of Amsterdam, Amsterdam, The Netherlands

☉ These authors contributed equally to this work.
‡ These authors also contributed equally to this work
* L.Meyaard@umcutrecht.nl

## Abstract

Signal inhibitory receptor on leukocytes-1 (SIRL-1) is an immune inhibitory receptor expressed on human myeloid cells. We previously showed that dendritic cell (DC)-driven Th17 cell differentiation of human naive CD4[+] T cells requires presence of neutrophils, which is inhibited by SIRL-1 ligation. VSTM1-v2 is a soluble isoform of SIRL-1, which was previously proposed to function as a Th17 polarizing cytokine. Here, we investigated the effect of VSTM1-v2 on DC-driven Th17 cell development. Neutrophils induced DC-driven Th17 cell differentiation, which was not enhanced by VSTM1-v2. Similarly, we found no effect of VSTM1-v2 on cytokine-driven Th17 cell development. Thus, our results do not support a role for VSTM1-v2 in Th17 cell differentiation.

## Introduction

Signal inhibitory receptor on leukocytes-1 (SIRL-1), encoded by the gene *VSTM1*, is an immune inhibitory receptor expressed on human granulocytes and monocytes [1–4]. We recently described that activated neutrophils shed the ectodomain of SIRL-1, thereby releasing soluble SIRL-1 (sSIRL-1) [5]. In addition, SIRL-1 has a soluble isoform named VSTM1-v2, which lacks the exon that encodes the transmembrane domain of SIRL-1 and is therefore predicted to give rise to a secreted protein [6].

VSTM1-v2 mRNA expression has been reported to be increased in peripheral blood mononuclear cells (PBMCs) of rheumatoid arthritis patients compared to controls, and to correlate positively with IL-17 mRNA expression [7]. Moreover, recombinant VSTM1-v2 has been shown to induce Th17 cell differentiation and activation, using purified CD4[+] T cells that were stimulated with a Th17 cell inducing cytokine cocktail and/or anti-CD3 and anti-CD28 stimulation [6], suggesting that VSTM1-v2 acts as a polarizing cytokine on CD4[+] T cells. We previously showed that DC-driven differentiation of human naive CD4[+] T cells into Th17 cells requires presence of neutrophils, which is inhibited by SIRL-1 ligation [8].

**Data Availability Statement:** All relevant data are within the paper and its Supporting Information files.

**Funding:** This work was supported by Amsterdam UMC, University of Amsterdam, and by a Vici grant

from the Netherlands Organization for Scientific Research (NWO, grant no. 91815608, www.nwo.nl). The funders had no role in study design, data collection and analysis, decision to publish, or preparation of the manuscript.

It is unclear whether VSTM1-v2 also affects DC-driven differentiation into Th17 cells. Here, we tested the effect of VSTM1-v2 on DC-driven and cytokine-driven Th17 cell development and assessed binding of VSTM1-v2 to leukocytes.

## Materials and methods

### Biological samples

Heparinized blood was obtained from healthy volunteers at the UMC Utrecht or Amsterdam UMC. Samples were collected in accordance with the Institutional Review Board of these institutes (METC 2015_074) and after written consent.

### Recombinant proteins

The cDNA sequence, cloning and production of recombinant His-tagged VSTM1-v2 and LAIR-1 ectodomain (sLAIR-1$^{ecto}$) has been previously described [5]. Endotoxins were removed with Triton X-114 (Sigma-Aldrich) and SM-2 beads (Bio-Rad) as previously described [9]. Absence of endotoxins (< 1EU/mL) was confirmed using the PyroGene$^{TM}$ Recombinant Factor C Assay (Lonza) according to manufacturer's instructions.

### DC-driven Th17 cell differentiation

Isolation and co-cultures of T cells, monocyte-derived DCs (moDCs) and neutrophils were performed as previously described [8]. In brief, following CD4$^+$ T-cell isolation, CD45RA$^+$ naive cells were separated from the CD45RO$^+$ memory T cells by positive selection of memory T cells using CD45RO-Phycoerythrin antibody (DAKO) and magnetically-labeled anti-PE beads (Miltenyi Biotec). Naive CD4$^+$ T cells were always more than 98% pure. Monocytes were differentiated into moDCs and harvested as immature DCs after 6 days. Co-cultures were initiated in 96-well *Candida albicans* hyphae-coated plates, with 50,000 DCs and 50,000 autologous CD4$^+$ naive or memory T cells, with or without 100 ng/mL VSTM1-v2, and for the naive T cell cultures, with or without 100,000 autologous freshly isolated neutrophils. Co-cultures were done in IMDM supplemented with 5% HI-HS (Lonza) and gentamycin (86 μg/mL, Duchefa Biochemie). After 4 days, cells were transferred to 48-well plates (Costar) and refreshed every 2 days with IMDM/5% HS medium containing 10 U/mL IL-2. At 10–12 days of culture, when cells were resting, they were restimulated for 5h with PMA (100 ng/mL), ionomycin (1 μg/mL), and brefeldin A (10 μg/mL) (all Sigma-Aldrich).

### Cytokine-driven Th17 cell differentiation

CD4$^+$ T cells were isolated from peripheral blood lymphocytes by negative selection using a CD4 T-cell isolation kit II and magnetic cell separation (Miltenyi Biotec). A total of 50,000 CD4$^+$ T cells were cultured in IMDM (Thermo Scientific, Gibco) supplemented with 10% FCS (Gibco) and stimulated with plate-bound anti-CD3 (16A9, 1 μg/mL, Sanquin) and soluble anti-CD28 (15E8, 1 μg/mL, Sanquin [10], in the presence of Th17 polarizing cytokines IL-1β (10 ng/mL), IL-6 (50 ng/mL), IL-23 (50 ng/mL), TGF-β (5 ng/mL) and TNF-α (10 ng/mL) and neutralizing antibodies anti-IFN-γ (10 μg/mL) and anti-IL-4 (10 μg/mL). IL-1β, IL-6 and TNF-α were purchased from Miltenyi Biotech, IL-23 and TGF-β from R&D Systems, anti-IFN-γ from U-CyTech and anti-IL-4 from BD Pharmingen. VSTM1-v2 was added at 10 or 100 ng/mL, or medium was added as control. After 4 days, cells were transferred to 24-well plates (Costar) and refreshed every 2 days with IMDM/10% FCS medium containing 10 U/mL recombinant human IL-2 (Novartis AG). After a total culture of 10–12 days, when cells were resting, they were re-stimulated with PMA, ionomycin, and brefeldin A as described above.

## Binding assays

Erythrocytes from full blood samples were lysed with Ammonium-Chloride-Potassium (ACK) lysis buffer (155 mM $NH_4Cl$, 10 mM $KHCO_3$, 0.1 mM EDTA in $ddH_2O$, pH 7.2–7.4). The remaining leukocytes were incubated with 10 or 50 μg/mL VSTM1-v2 or sLAIR-1ecto for 90 minutes at 4˚C, after which the cells were stained with anti-Penta-His-AF647 (1 μg/mL; Qiagen) for 30 minutes at 4˚C. Between all steps, cells were extensively washed with PBS containing 1% BSA and 0.01% $NaN_3$. Cells were analyzed by flow cytometry.

## Flow cytometry

Cells were acquired on a FACS Canto II (BD Biosciences). Lymphocytes, monocytes and granulocytes were distinguished based on forward scatter (FSC) and sideward scatter (SSC) (Fig 1A). For intracellular IL-17 staining, restimulated T cells were washed with Saponin (Sigma-Aldrich) in PBS-0.5% w/v BSA-0.05% v/v azide and stained with anti-IL-17A-eFluor660. Gating was performed as previously described [8]. Flow cytometry analysis was performed using FlowJo software (Treestar, Ashland, OR).

## Statistics

Statistical analyses were performed using GraphPad Prism Software, La Jolla, CA, USA, version 8.3.0 for Windows. For each graph, the statistical test and number of biological replicates (n) are described in the figure legends. P values of 0.05 or less were considered significant.

## Results

To test the effect of VSTM1-v2 on DC-driven Th17 cell differentiation, we used our previously described co-culture system [8] of naive CD4+ T cells, *Candida albicans*-activated moDCs, and neutrophils (See Fig 1A, upper panel, for a schematic overview). *C. albicans* is a potent Th17 inducing pathogen in DC-driven T cell outgrowth, and patients with genetic errors in IL-17 expression suffer from chronic mucocutaneous candidiasis [11]. In line with our previous data, presence of neutrophils promoted the differentiation of naive CD4+ T cells into IL-17+ T cells (Fig 1B and 1C). However, VSTM1-v2 did not consistently change the percentage of IL-17+ T cells in these cultures (Fig 1C, left panel).

We previously showed that *C. albicans*-activated moDCs also enhance Th17 cell activation of memory CD4+ T cells [8]. Therefore, we subsequently tested the effect of VSTM1-v2 on Th17 cell activation from memory CD4+ T cells (see Fig 1A, lower panel for the co-culture set-up). Similar to our previous findings, co-culture of memory CD4+ T cells with *C. albicans*-activated moDCs induced up to 30% of IL-17+ cells, but this was not increased by addition of VSTM1-v2 (Fig 1C, right panel). Taken together, we found no effect of VSTM1-v2 on DC-driven Th17 cell differentiation nor activation.

In the co-culture systems, a direct effect of VSTM1-v2 on Th17 cell development may be blocked if moDCs or granulocytes compete with CD4+ T cells for VSTM1-v2 binding. To rule this out, we assessed binding of VSTM1-v2 to granulocytes, lymphocytes and monocytes in erythrocyte-lysed whole blood (See S1 Fig panel A for the gating strategy). As a negative control, sLAIR-1ecto was used, which was produced in the same way as VSTM1-v2 and has 30% sequence identity with VSTM1-v2. For detection, fluorescently labeled anti-His was used. In comparison to the staining by anti-His alone, we observed no increased binding of VSTM1-v2 to lymphocytes, monocytes, or granulocytes (S1 Fig panel B-E), indicating that VSTM1-v2 does not bind directly to human leukocytes.

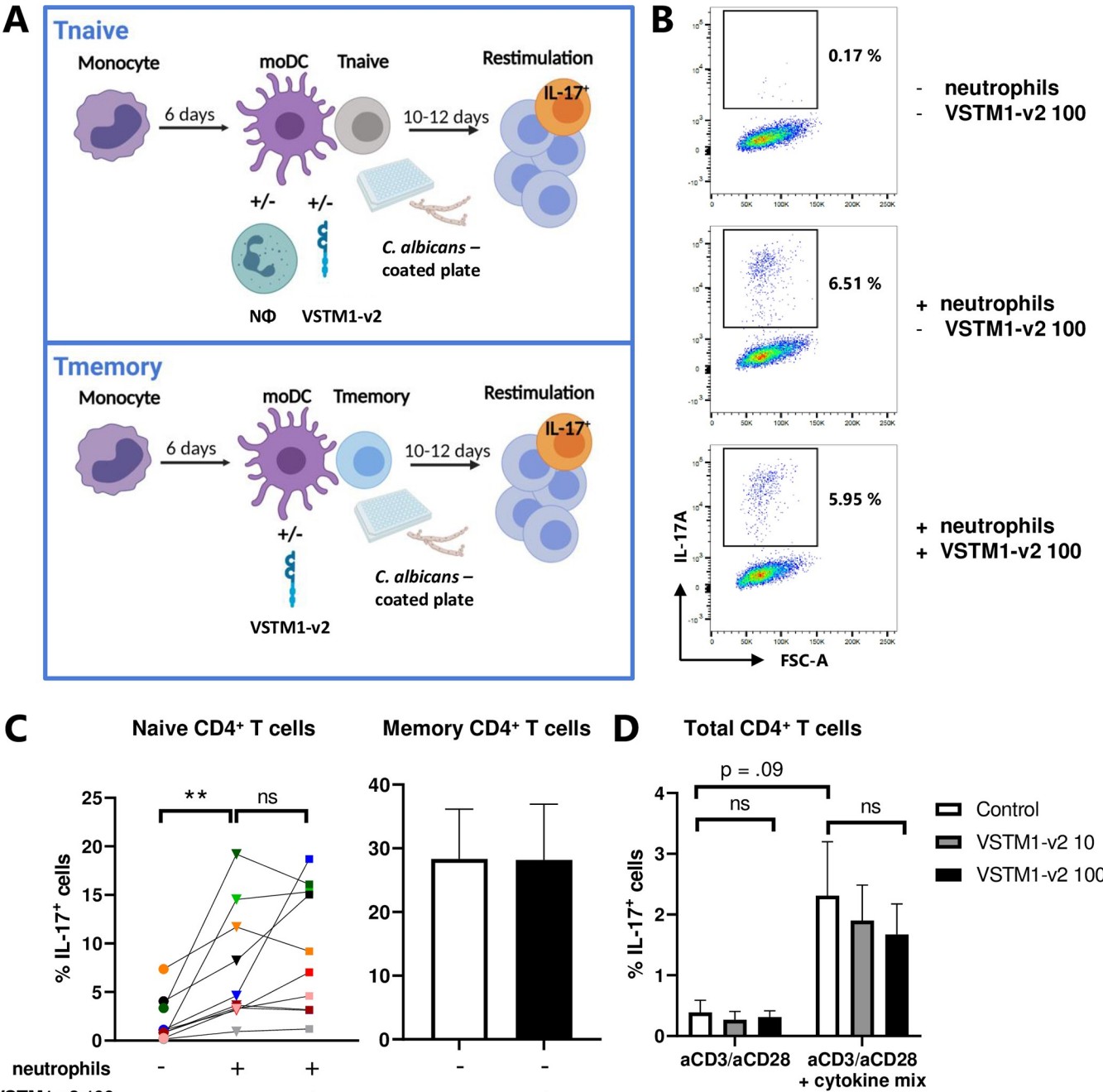

**Fig 1. VSTM1-v2 does not enhance Th17 cell differentiation and activation.** CD4$^+$ T cells were stimulated by *C. albicans*-activated moDCs, with or without autologous neutrophils (A-C), or with antibodies and a polarizing cytokine mix (D), with or without addition of 10 or 100 ng/mL VSTM1-v2. Intracellular IL-17 expression was determined by flow cytometry. **(A)** Schematic representation of the co-culture system of moDC-driven Th17 cell differentiation and activation. **(B)** Representative dot plots of the percentage of IL-17$^+$ cells after co-culture of naive CD4$^+$ T cells. The y-axis indicates the fluorescence intensity of the IL-17A staining, while the x-axis indicates the forward scatter (FSC). **(C)** The percentage of IL-17$^+$ cells after co-culture of naive CD4$^+$ T cells (n = 10), each donor represented by a different color, or memory CD4$^+$ T cells (n = 3; mean ± SD). **(D)** The percentage of IL-17$^+$ cells after stimulation of total CD4$^+$ T cells with anti-CD3, anti-CD28, and a Th17 polarizing mix, mean ± SD, n = 3. Statistical significance was determined using a Friedman test with Dunn's correction (C, D). ** p < .01, ns = not significant. Figure A was made using biorender.com.

Finally, we repeated the same set-up as Guo *et al.* used to show that VSTM1-v2 induced Th17 cell differentiation, by activating total CD4[+] T cells with anti-CD3, anti-CD28, and a Th17 polarizing cytokine mix [6]. The Th17 polarizing mix increased the percentage of IL-17[+] cells, but there was no effect of VSTM1-v2 (Fig 1D). Thus, our data do not support a role for VSTM1-v2 in the differentiation or activation of Th17 cells.

## Discussion

We confirmed our previous observations that neutrophils are required for DC-driven differentiation of naive CD4[+] T cells into Th17 cells [8]. Mechanistically, we explained this effect by release of elastase from activated neutrophils, which cleaved DC-derived CXCL8 into a Th17-cell inducing agent. SIRL-1 ligation inhibited Th17 cell development, most likely by suppression of neutrophil degranulation and thereby the release of neutrophil elastase [8]. Based on those findings, we hypothesized that VSTM1-v2 may revert the suppressive effect of SIRL-1 on Th17 cell development by competing with SIRL-1 for binding to its ligands S100 and LL-37 [12,13], which are both neutrophil-derived proteins. Of note, we were not able to detect direct binding between SIRL-1 and its ligands [12,13], possibly due to low affinity or the requirement of an additional binding partner. Therefore we could technically not assess whether VSTM1-v2 competes with this interaction. More importantly, we did not find any effect of VSTM1-v2 on Th17 cell differentiation and activation, unlike the results by Guo *et al.* [6].

We neither observed effects of VSTM1-v2 on DC-driven nor cytokine-driven T cell outgrowth. It is unclear what caused these differential results with the study by Guo *et al.*. Purified VSTM1-v2 had a similar molecular weight as in the study by Guo *et al.*, as assessed by SDS-PAGE [5]. We also used a similar Th17 polarizing mix and the same VSTM1-v2 concentrations, but we cultured the cells for 10–12 days before staining them for IL-17 expression, whereas Guo *et al.* found that IL-17 release by total CD4[+] T cells was already increased after 72 h stimulation [6]. This could indicate that pre-existing Th17 cells within the memory CD4[+] T cell population were stimulated to produce more IL-17, rather than amplifying the percentage of IL-17[+] cells. However, how VSTM1-v2 would activate IL-17 production by T cells is unclear, since we did not observe any binding to lymphocytes (S1 Fig).

As of now, endogenous expression of VSTM1-v2 has only been shown on mRNA level [6,7], but not as a protein. We recently described that sSIRL-1 protein is released by ectodomain shedding [5]. Using an in-house developed ELISA, we found sSIRL-1 concentration increased in COVID-19 and RSV bronchiolitis patients. Notably, this ELISA can also recognize recombinant VSTM1-v2, and VSTM1-v2 may therefore contribute to the total amount of sSIRL-1 that was found. In future, VSTM1-v2 specific antibodies will allow investigation of endogenous VSTM1-v2 protein expression.

In conclusion, our results do not support a role for VSTM1-v2 in Th17 cell differentiation. Further research is required to elucidate the expression of VSTM1-v2 protein and its functional implications.

## Supporting information

**S1 Fig. VSTM1-v2 does not bind to leukocytes.** Erythrocyte-lysed whole blood was incubated with His-tagged VSTM1-v2 or sLAIR-1ecto (10–50 µg/mL), followed by detection with anti-His-AF647. **(A)** Lymphocytes (lower population), monocytes (middle population) and granulocytes (upper population) were gated based on forward scatter (FSC) and sideward scatter (SSC), followed by gating on single cells and AF-647+ cells. **(B)** Shown are representative dot plots of lymphocytes stained with 50 µg/mL VSTM1-v2 and/or anti-His-AF647. **(C-E)** The graphs show the quantification of the percentage of AF647+ cells of lymphocytes (C),

monocytes (D), or granulocytes (E). Data are shown as mean ± SD of two donors.
(TIF)

## Acknowledgments

We thank all members of the Meyaard lab and the de Jong lab for valuable discussions on the manuscript.

## Author Contributions

**Conceptualization:** Helen J. von Richthofen, Florianne M. J. Hafkamp, Esther C. de Jong, Linde Meyaard.

**Formal analysis:** Helen J. von Richthofen, Florianne M. J. Hafkamp.

**Funding acquisition:** Esther C. de Jong, Linde Meyaard.

**Investigation:** Helen J. von Richthofen, Florianne M. J. Hafkamp, Anouk van Haperen.

**Methodology:** Helen J. von Richthofen, Florianne M. J. Hafkamp, Esther C. de Jong, Linde Meyaard.

**Supervision:** Esther C. de Jong, Linde Meyaard.

**Visualization:** Helen J. von Richthofen, Florianne M. J. Hafkamp.

**Writing – original draft:** Helen J. von Richthofen, Florianne M. J. Hafkamp.

**Writing – review & editing:** Esther C. de Jong, Linde Meyaard.

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
