## [Decision Letter · Decision Letter 0]

10 Nov 2022

PONE-D-22-25685VSTM1-v2 does not drive human Th17 cell differentiation: a replication study.PLOS ONE

Dear Dr. Meyaard,

Thank you for submitting your manuscript to PLOS ONE. After careful consideration, we feel that it has merit but does not fully meet PLOS ONE’s publication criteria as it currently stands. Therefore, we invite you to submit a revised version of the manuscript that addresses the points raised during the review process.

We look forward to receiving your revised manuscript.

Kind regards,

Sumit Kumar Hira, Ph.D.

Academic Editor

PLOS ONE

Journal Requirements:

"This work was supported by Amsterdam UMC, University of Amsterdam, and by a Vici grant from the Netherlands Organization for Scientific Research (NWO, grant no. 91815608)"

"This work was supported by Amsterdam UMC, University of Amsterdam, and by a Vici grant from the Netherlands Organization for Scientific Research (NWO, grant no. 91815608, www.nwo.nl). The funders had no role in study design, data collection and analysis, decision to publish, or preparation of the manuscript."

Reviewers' comments:

Reviewer's Responses to Questions

**Comments to the Author**

1. Is the manuscript technically sound, and do the data support the conclusions?

Reviewer #1: Yes

Reviewer #2: Partly

2. Has the statistical analysis been performed appropriately and rigorously? 

Reviewer #1: Yes

Reviewer #2: Yes

3. Have the authors made all data underlying the findings in their manuscript fully available?

Reviewer #1: Yes

Reviewer #2: Yes

4. Is the manuscript presented in an intelligible fashion and written in standard English?

Reviewer #1: Yes

Reviewer #2: Yes

5. Review Comments to the Author

Reviewer #1: For figure 1B, n=7 seems too low for providing conclusive results. few more samples should be included ( n=20 fir example).

Representative Dot plot for IL17+ cells should be included at least as supporting figures.

ELISA of IL17 from Naive and memory T cells after treatment ( for cytokine induced Th17 differentiation ) could be considered

Reviewer #2: The manuscript provides a controversial opinion on the role of VSTM1-v2 in human Th17 cell differentiation against a previous study. To draw a clear conclusion, the author should supply additional data to answer the following questions:

1) The VSTM1-v2 expression construct has not be verified by sequencing therefore, the protein generated using this construct may not be correct.

2) It is not clear whether VSTM1-v2 interacts with a ligand used by the full length SIRL-1.

3) Since purified VSTM1-v2 may lose function, overexpression of this protein may be more feasible.

6. PLOS authors have the option to publish the peer review history of their article (what does this mean?). If published, this will include your full peer review and any attached files.

Reviewer #1: **Yes: **Sankar Bhattacharyya, Ph. D, Associate Professor, Department of Zoology, Sidho Kanho Birsha University, India.

Reviewer #2: No

---

## [Author Response · Author response to Decision Letter 0]

17 Jan 2023

Manuscript reference number: PONE-D-22-25685

Dear Editor,

We thank you for the evaluation of our manuscript entitled “VSTM1-v2 does not drive human Th17 cell differentiation: a replication study”.

We have addressed the reviewers’ comments and performed the requested experiment to increase the number of replicates. You can find our point-by-point reply to the reviewers’ comments below. Changes in the original manuscript are indicated as tracked changes.

We hope that, with these changes, you will find the manuscript acceptable for publication in PLOS ONE.

Our funding statement is as follows:

“This work was supported by Amsterdam UMC, University of Amsterdam, and by a Vici grant from the Netherlands Organization for Scientific Research (NWO, grant no. 91815608, www.nwo.nl ). The funders had no role in study design, data collection and analysis, decision to publish, or preparation of the manuscript."

Yours sincerely, 

Helen von Richthofen

Linde Meyaard

Reviewer #1: For figure 1B, n=7 seems too low for providing conclusive results. few more samples should be included ( n=20 fir example).

Representative Dot plot for IL17+ cells should be included at least as supporting figures.

ELISA of IL17 from Naive and memory T cells after treatment ( for cytokine induced Th17 differentiation ) could be considered

Reply:

As requested by reviewer #1, we increased the number of replicates to strengthen the conclusion that VSTM1-v2 does not enhance DC-driven Th17 cell development from human naive T cells. Addition up to n=20 was not feasible as per experiment this takes approximately three weeks, but we did add n=3 to the figure with now a total of n=10. Statistical analysis still shows that no significant induction of Th17 cells occurs by VSTM1-v2. Furthermore, we included representative dot plots for IL-17+ cells as panel B in Figure 1. 

We thank reviewer #1 for the suggestion to consider an ELISA to measure IL-17 in supernatant as a measure for Th17 cell differentiation. In our previous publication by Souwer et al. [1], we also assessed DC-driven Th17 differentiation with or without neutrophils, and we measured a similar fold induction in % IL-17+ cells versus IL-17 concentration in supernatant. Thus, these methods seem equally sensitive. However, measuring the % of IL-17+ cells also gives single cell information as opposed to using ELISA, which is why we chose this method.

Reviewer #2: The manuscript provides a controversial opinion on the role of VSTM1-v2 in human Th17 cell differentiation against a previous study. To draw a clear conclusion, the author should supply additional data to answer the following questions:

1) The VSTM1-v2 expression construct has not be verified by sequencing therefore, the protein generated using this construct may not be correct.

Reply: We did sequence the VSTM1-v2 construct as part of our standard cloning procedure, and confirmed that the sequence is correct. The sequence can be found in our recent publication in J Immunol (Supplemental Figure 1) [2].

2) It is not clear whether VSTM1-v2 interacts with a ligand used by the full length SIRL-1.

Reply: We did not detect direct binding between SIRL-1 and its ligands (S100 proteins, LL-37 and phenol soluble modulins) [3, 4], possibly due to low affinity interactions or the requirement of an additional binding partner. As a consequence, we were unfortunately not able to technically assess whether VSTM1-v2 competes with the binding between SIRL-1 and its ligands. We now clarified this in sentence 168-170 of the manuscript.

3) Since purified VSTM1-v2 may lose function, overexpression of this protein may be more feasible.

Reply: We appreciate the suggestion of the reviewer, but we purposefully chose to use purified recombinant VSTM1-v2 for several reasons; 1) we aimed to replicate the study by Guo et al., in which purified VSTM1-v2 was used; 2) purification of recombinant proteins by His-tag is an approved and commonly used method to obtain functional recombinant proteins [5], and 3) we made use of primary cells in this system, which are technically challenging to transfect for overexpression.

References

1. Souwer Y, Groot Kormelink T, Taanman-Kueter EW, Muller FJ, van Capel TMM, Varga DV, et al. Human TH17 cell development requires processing of dendritic cell-derived CXCL8 by neutrophil elastase. The Journal of allergy and clinical immunology. 2018;141(6):2286-9 e5.

2. von Richthofen HJ, Westerlaken GHA, Gollnast D, Besteman S, Delemarre EM, Rodenburg K, et al. Soluble Signal Inhibitory Receptor on Leukocytes-1 Is Released from Activated Neutrophils by Proteinase 3 Cleavage. J Immunol. 2023.

3. Rumpret M, von Richthofen HJ, van der Linden M, Westerlaken GHA, Talavera Ormeno C, Low TY, et al. Recognition of S100 proteins by Signal Inhibitory Receptor on Leukocytes-1 negatively regulates human neutrophils. Eur J Immunol. 2021;51(9):2210-7.

4. Rumpret M, von Richthofen HJ, van der Linden M, Westerlaken GHA, Talavera Ormeno C, van Strijp JAG, et al. Signal inhibitory receptor on leukocytes-1 recognizes bacterial and endogenous amphipathic alpha-helical peptides. FASEB J. 2021;35(10):e21875.

5. Spriestersbach A, Kubicek J, Schafer F, Block H, Maertens B. Purification of His-Tagged Proteins. Methods Enzymol. 2015;559:1-15.

---

## [Decision Letter · Decision Letter 1]

16 Feb 2023

PONE-D-22-25685R1

VSTM1-v2 does not drive human Th17 cell differentiation: a replication study.

PLOS ONE

Dear Dr. Meyaard,

Thank you for submitting your manuscript to PLOS ONE. After careful consideration, we feel that it has merit but does not fully meet PLOS ONE’s publication criteria as it currently stands. Therefore, we invite you to submit a revised version of the manuscript that addresses the points raised during the review process.

We look forward to receiving your revised manuscript.

Kind regards,

Sumit Kumar Hira, Ph.D.

Academic Editor

PLOS ONE

Journal Requirements:

Reviewers' comments:

Reviewer's Responses to Questions

**Comments to the Author**

1. If the authors have adequately addressed your comments raised in a previous round of review and you feel that this manuscript is now acceptable for publication, you may indicate that here to bypass the “Comments to the Author” section, enter your conflict of interest statement in the “Confidential to Editor” section, and submit your "Accept" recommendation.

Reviewer #1: (No Response)

Reviewer #2: All comments have been addressed

2. Is the manuscript technically sound, and do the data support the conclusions?

Reviewer #1: Yes

Reviewer #2: No

3. Has the statistical analysis been performed appropriately and rigorously? 

Reviewer #1: Yes

Reviewer #2: Yes

4. Have the authors made all data underlying the findings in their manuscript fully available?

Reviewer #1: Yes

Reviewer #2: Yes

5. Is the manuscript presented in an intelligible fashion and written in standard English?

Reviewer #1: Yes

Reviewer #2: Yes

6. Review Comments to the Author

Reviewer #1: (No Response)

Reviewer #2: Although purified proteins usually have activity that was stated in literature, it is not necessary true in every lab and for every protein. The authors produced the protein in primary cells, the supernatants or the primary cells themselves carrying the transfected expression construct would be a useful resource containing the protein that more likely has activity.

7. PLOS authors have the option to publish the peer review history of their article (what does this mean?). If published, this will include your full peer review and any attached files.

Reviewer #1: **Yes: **Sankar Bhattacharyya

Reviewer #2: No

---

## [Author Response · Author response to Decision Letter 1]

27 Feb 2023

Manuscript reference number: PONE-D-22-25685

Dear Dr Sumit Kumar Hira,

Thank you for your evaluation of the revision of our manuscript entitled “VSTM1-v2 does not drive human Th17 cell differentiation: a replication study” for PLOS ONE. 

Both reviewers now indicate that all their comments were addressed in the revised manuscript, yet your recommendation is to revise the manuscript again. We assume this is because of reviewer #2’s comment to question 6, suggesting to use supernatants of VSTM1-v2-transfected cells instead of purified VSTM1-v2 for our experiments.

In the first round of review, reviewer #2 also raised this concern. We replied to this comment in the following way: “… we purposefully chose to use purified recombinant VSTM1-v2 for several reasons; 1) we aimed to replicate the study by Guo et al., in which purified VSTM1-v2 was used; 2) purification of recombinant proteins by His-tag is an approved and commonly used method to obtain functional recombinant proteins, and 3) we made use of primary cells in this system, which are technically challenging to transfect for overexpression.”

These arguments still stand, we purposely chose to use purification of His-tagged recombinant VSTM1-v2 with a Ni-Sepharose column in order to replicate the method used by Guo et al. Additionally, cell culture supernatant contains a plethora of compounds which may have indirect effects on the co-culture system that we use.

We characterized VSTM1-v2 by SDS-PAGE and Western blot after purification (von Richthofen et al., 2023). VSTM1-v2 had a molecular weight of approximately 50 kDa, and deglycosylated VSTM1-v2 had a molecular weight of 37 kDa. This is in accordance with the findings by Guo et al.. Moreover, we could detect native VSTM1-v2 with anti-SIRL-1 antibodies (two clones), which is to be expected since VSTM1-v2 is a splice isoform of SIRL-1. This suggests that purification of VSTM1-v2 resulted in a viable protein. We did not describe the characterization of VSTM1-v2 protein in the current manuscript, since we already described it in our earlier publication. However, to avoid confusion, we now added the following sentence to the discussion, after sentence 175: “Purified VSTM1-v2 had a similar molecular weight as in the study by Guo et al., as assessed by SDS-PAGE (von Richthofen et al., 2023). “

Taken together, we conclude that using supernatants of VSTM1-v2 transfected cells rather than purified VSTM1-v2 will not improve the quality of the manuscript. We therefore hope that with this addition the manuscript is acceptable for publication in PLOS ONE .

Yours sincerely, 

Helen von Richthofen

Linde Meyaard

von Richthofen, H. J., Westerlaken, G. H. A., Gollnast, D., Besteman, S., Delemarre, E. M., Rodenburg, K., . . . Meyaard, L. (2023). Soluble Signal Inhibitory Receptor on Leukocytes-1 Is Released from Activated Neutrophils by Proteinase 3 Cleavage. Journal of Immunology, 210(4), 389-397. doi:10.4049/jimmunol.2200169

---

## [Editor Report · Decision Letter 2]

30 Mar 2023

VSTM1-v2 does not drive human Th17 cell differentiation: a replication study.

PONE-D-22-25685R2

Dear Dr. Meyaard,

We’re pleased to inform you that your manuscript has been judged scientifically suitable for publication and will be formally accepted for publication once it meets all outstanding technical requirements.

Kind regards,

Sumit Kumar Hira, Ph.D.

Academic Editor

PLOS ONE
---

## [Editor Report · Acceptance letter]

6 Apr 2023

PONE-D-22-25685R2 

VSTM1-v2 does not drive human Th17 cell differentiation: a replication study. 

Dear Dr. Meyaard:

I'm pleased to inform you that your manuscript has been deemed suitable for publication in PLOS ONE. Congratulations! Your manuscript is now with our production department. 

Kind regards, 

on behalf of

Dr. Sumit Kumar Hira 

Academic Editor

PLOS ONE